# Measuring the reliability of MCMC inference with bidirectional Monte Carlo

**Roger B. Grosse**
Department of Computer Science
University of Toronto

**Siddharth Ancha**
Department of Computer Science
University of Toronto

**Daniel M. Roy**
Department of Statistics
University of Toronto

## Abstract

Markov chain Monte Carlo (MCMC) is one of the main workhorses of probabilistic inference, but it is notoriously hard to measure the quality of approximate posterior samples. This challenge is particularly salient in black box inference methods, which can hide details and obscure inference failures. In this work, we extend the recently introduced bidirectional Monte Carlo [GGA15] technique to evaluate MCMC-based posterior inference algorithms. By running annealed importance sampling (AIS) chains both from prior to posterior and vice versa on simulated data, we upper bound in expectation the symmetrized KL divergence between the true posterior distribution and the distribution of approximate samples. We integrate our method into two probabilistic programming languages, WebPPL [GS] and Stan [CGHL+ p], and validate it on several models and datasets. As an example of how our method be used to guide the design of inference algorithms, we apply it to study the effectiveness of different model representations in WebPPL and Stan.

## 1 Introduction

Markov chain Monte Carlo (MCMC) is one of the most important classes of probabilistic inference methods and underlies a variety of approaches to automatic inference [e.g. LTBS00; GMRB+08; GS; CGHL+ p]. Despite its widespread use, it is still difficult to rigorously validate the effectiveness of an MCMC inference algorithm. There are various heuristics for diagnosing convergence, but reliable quantitative measures are hard to find. This creates difficulties both for end users of automatic inference systems and for experienced researchers who develop models and algorithms.

In this paper, we extend the recently proposed bidirectional Monte Carlo (BDMC) [GGA15] method to evaluate certain kinds of MCMC-based inference algorithms by bounding the symmetrized KL divergence (Jeffreys divergence) between the distribution of approximate samples and the true posterior distribution. Specifically, our method is applicable to algorithms which can be viewed as importance sampling over an extended state space, such as annealed importance sampling (AIS; [Nea01]) or sequential Monte Carlo (SMC; [MDJ06]). BDMC was proposed as a method for accurately estimating the log marginal likelihood (log-ML) on simulated data by sandwiching the true value between stochastic upper and lower bounds which converge in the limit of infinite computation. These log-likelihood values were used to benchmark marginal likelihood estimators. We show that it can also be used to measure the accuracy of approximate posterior samples obtained from algorithms like AIS or SMC. More precisely, we refine the analysis of [GGA15] to derive an estimator which upper bounds in expectation the Jeffreys divergence between the distribution of approximate samples and the true posterior distribution. We show that this upper bound is quite accurate on some toy distributions for which both the true Jeffreys divergence and the upper bound can be computed exactly. We refer to our method of bounding the Jeffreys divergence by sandwiching the log-ML as Bounding Divergences with REverse Annealing (BREAD).

While our method is only directly applicable to certain algorithms such as AIS or SMC, these algorithms involve many of the same design choices as traditional MCMC methods, such as the choice of model representation (e.g. whether to collapse out certain variables), or the choice of MCMC transition operators. Therefore, the ability to evaluate AIS-based inference should also yield insights which inform the design of MCMC inference algorithms more broadly.

One additional hurdle must be overcome to use BREAD to evaluate posterior inference: the method yields rigorous bounds only for simulated data because it requires an exact posterior sample. One would like to be sure that the results on simulated data accurately reflect the accuracy of posterior inference on the real-world data of interest. We present a protocol for using BREAD to diagnose inference quality on real-world data. Specifically, we infer hyperparameters on the real data, simulate data from those hyperparameters, measure inference quality on the simulated data, and validate the consistency of the inference algorithm's behavior between the real and simulated data. (This protocol is somewhat similar in spirit to the parametric bootstrap [ET98].)

We integrate BREAD into the tool chains of two probabilistic programming languages: WebPPL [GS] and Stan [CGHL+ p]. Both probabilistic programming systems can be used as automatic inference software packages, where the user provides a program specifying a joint probabilistic model over observed and unobserved quantities. In principle, probabilistic programming has the potential to put the power of sophisticated probabilistic modeling and efficient statistical inference into the hands of non-experts, but realizing this vision is challenging because it is difficult for a non-expert user to judge the reliability of results produced by black-box inference. We believe BREAD provides a rigorous, general, and automatic procedure for monitoring the quality of posterior inference, so that the user of a probabilistic programming language can have confidence in the accuracy of the results. Our approach to evaluating probabilistic programming inference is closely related to independent work [CTM16] that is also based on the ideas of BDMC. We discuss the relationships between both methods in Section 4.

In summary, this work includes four main technical contributions. First, we show that BDMC yields an estimator which upper bounds in expectation the Jeffreys divergence of approximate samples from the true posterior. Second, we present a technique for exactly computing both the true Jeffreys divergence and the upper bound on small examples, and show that the upper bound is often a good match in practice. Third, we propose a protocol for using BDMC to evaluate the accuracy of approximate inference on real-world datasets. Finally, we extend both WebPPL and Stan to implement BREAD, and validate BREAD on a variety of probabilistic models in both frameworks. As an example of how BREAD can be used to guide modeling and algorithmic decisions, we use it to analyze the effectiveness of different representations of a matrix factorization model in both WebPPL and Stan.

## 2 Background

### 2.1 Annealed Importance Sampling

Annealed importance sampling (AIS; [Nea01]) is a Monte Carlo algorithm commonly used to estimate (ratios of) normalizing constants. More carefully, fix a sequence of $T$ distributions $p_1, \ldots, p_T$, with $p_t(\mathbf{x}) = f_t(\mathbf{x})/\mathcal{Z}_t$. The final distribution in the sequence, $p_T$, is called the *target* distribution; the first distribution, $p_1$, is called the *initial* distribution. It is required that one can obtain one or more exact samples from $p_1$.[1] Given a sequence of reversible MCMC transition operators $\mathcal{T}_1, \ldots, \mathcal{T}_T$, where $\mathcal{T}_t$ leaves $p_t$ invariant, AIS produces a (nonnegative) unbiased estimate of $\mathcal{Z}_T/\mathcal{Z}_1$ as follows: first, we sample a random initial state $\mathbf{x}_1$ from $p_1$ and set the initial weight $w_1 = 1$. For every stage $t \geq 2$ we update the weight $w$ and sample the state $\mathbf{x}_t$ according to

$$w_t \leftarrow w_{t-1} \frac{f_t(\mathbf{x}_{t-1})}{f_{t-1}(\mathbf{x}_{t-1})} \qquad\qquad \mathbf{x}_t \leftarrow \text{sample from } \mathcal{T}_t\left(\mathbf{x} \,|\, \mathbf{x}_{t-1}\right). \qquad (1)$$

Neal [Nea01] justified AIS by showing that it is a simple importance sampler over an extended state space (see Appendix A for a derivation in our notation). From this analysis, it follows that the weight $w_T$ is an unbiased estimate of the ratio $\mathcal{Z}_T/\mathcal{Z}_1$. Two trivial facts are worth highlighting: when $\mathcal{Z}_1$

is known, $\mathcal{Z}_1 w_T$ is an unbiased estimate of $\mathcal{Z}_T$, and when $\mathcal{Z}_T$ is known, $w_T / \mathcal{Z}_T$ is an unbiased estimate of $1/\mathcal{Z}_1$. In practice, it is common to repeat the AIS procedure to produce $K$ independent estimates and combine these by simple averaging to reduce the variance of the overall estimate.

In most applications of AIS, the normalization constant $\mathcal{Z}_T$ for the target distribution $p_T$ is the focus of attention, and the initial distribution $p_1$ is chosen to have a known normalization constant $\mathcal{Z}_1$. Any sequence of intermediate distributions satisfying a mild domination criterion suffices to produce a valid estimate, but in typical applications, the intermediate distributions are simply defined to be geometric averages $f_t(\mathbf{x}) = f_1(\mathbf{x})^{1-\beta_t} f_T(\mathbf{x})^{\beta_t}$, where the $\beta_t$ are monotonically increasing parameters with $\beta_1 = 0$ and $\beta_T = 1$. (An alternative approach is to average moments [GMS13].)

In the setting of Bayesian posterior inference over parameters $\boldsymbol{\theta}$ and latent variables $\mathbf{z}$ given some fixed observation $\mathbf{y}$, we take $f_1(\boldsymbol{\theta}, \mathbf{z}) = p(\boldsymbol{\theta}, \mathbf{z})$ to be the prior distribution (hence $\mathcal{Z}_1 = 1$), and we take $f_T(\boldsymbol{\theta}, \mathbf{z}) = p(\boldsymbol{\theta}, \mathbf{z}, \mathbf{y}) = p(\boldsymbol{\theta}, \mathbf{z}) \, p(\mathbf{y} | \boldsymbol{\theta}, \mathbf{z})$. This can be viewed as the unnormalized posterior distribution, whose normalizing constant $\mathcal{Z}_T = p(\mathbf{y})$ is the marginal likelihood. Using geometric averaging, the intermediate distributions are then

$$f_t(\boldsymbol{\theta}, \mathbf{z}) = p(\boldsymbol{\theta}, \mathbf{z}) \, p(\mathbf{y} | \boldsymbol{\theta}, \mathbf{z})^{\beta_t}. \tag{2}$$

In addition to moment averaging, reasonable intermediate distributions can be produced in the Bayesian inference setting by conditioning on a sequence of increasing subsets of data; this insight relates AIS to the seemingly different class of sequential Monte Carlo (SMC) methods [MDJ06].

## 2.2 Stochastic lower bounds on the log partition function ratio

AIS produces a nonnegative unbiased estimate $\hat{\mathcal{R}}$ of the ratio $\mathcal{R} = \mathcal{Z}_T / \mathcal{Z}_1$ of partition functions. Unfortunately, because such ratios often vary across many orders of magnitude, it frequently happens that $\hat{\mathcal{R}}$ underestimates $\mathcal{R}$ with overwhelming probability, while occasionally taking extremely large values. Correspondingly, the variance may be extremely large, or even infinite.

For these reasons, it is more meaningful to estimate $\log \mathcal{R}$. Unfortunately, the logarithm of a nonnegative unbiased estimate (such as the AIS estimate) is, in general, a biased estimator of the log estimand. More carefully, let $\hat{A}$ be a nonnegative unbiased estimator for $A = \mathbb{E}[\hat{A}]$. Then, by Jensen's inequality, $\mathbb{E}[\log \hat{A}] \le \log \mathbb{E}[\hat{A}] = \log A$, and so $\log \hat{A}$ is a lower bound on $\log A$ in expectation. The estimator $\log \hat{A}$ satisfies another important property: by Markov's inequality for nonnegative random variables, $\Pr(\log \hat{A} > \log A + b) < e^{-b}$, and so $\log \hat{A}$ is extremely unlikely to overestimate $\log A$ by any appreciable number of nats. These observations motivate the following definition [BGS15]: a *stochastic lower bound* on $X$ is an estimator $\hat{X}$ satisfying $\mathbb{E}[\hat{X}] \le X$ and $\Pr(\hat{X} > X + b) < e^{-b}$. Stochastic upper bounds are defined analogously. The above analysis shows that $\log \hat{A}$ is a stochastic lower bound on $\log A$ when $\hat{A}$ is a nonnegative unbiased estimate of $A$, and, in particular, $\log \hat{\mathcal{R}}$ is a stochastic lower bound on $\log \mathcal{R}$. (It is possible to strengthen the tail bound by combining multiple samples [GBD07].)

## 2.3 Reverse AIS and Bidirectional Monte Carlo

Upper and lower bounds are most useful in combination, as one can then sandwich the true value. As described above, AIS produces a stochastic lower bound on the ratio $\mathcal{R}$; many other algorithms do as well. Upper bounds are more challenging to obtain. The key insight behind bidirectional Monte Carlo (BDMC; [GGA15]) is that, *provided one has an exact sample from the target distribution $p_T$*, one can run AIS *in reverse* to produce a stochastic lower bound on $\log \mathcal{R}_{\mathrm{rev}} = \log \mathcal{Z}_1 / \mathcal{Z}_T$, and therefore a stochastic *upper* bound on $\log \mathcal{R} = -\log \mathcal{R}_{\mathrm{rev}}$. (In fact, BDMC is a more general framework which allows a variety of partition function estimators, but we focus on AIS for pedagogical purposes.)

More carefully, for $t = 1, \ldots, T$, define $\tilde{p}_t = p_{T-t+1}$ and $\tilde{\mathcal{T}}_t = \mathcal{T}_{T-t+1}$. Then $\tilde{p}_1$ corresponds to our original target distribution $p_T$ and $\tilde{p}_T$ corresponds to our original initial distribution $p_1$. As before, $\tilde{\mathcal{T}}_t$ leaves $\tilde{p}_t$ invariant. Consider the estimate produced by AIS on the sequence of distributions $\tilde{p}_1, \ldots, \tilde{p}_T$ and corresponding MCMC transition operators $\tilde{\mathcal{T}}_1, \ldots, \tilde{\mathcal{T}}_T$. (In this case, the forward chain of AIS corresponds to the reverse chain described in Section 2.1.) The resulting estimate $\hat{\mathcal{R}}_{\mathrm{rev}}$ is a nonnegative unbiased estimator of $\mathcal{R}_{\mathrm{rev}}$. It follows that $\log \hat{\mathcal{R}}_{\mathrm{rev}}$ is a stochastic lower bound on $\log \mathcal{R}_{\mathrm{rev}}$, and therefore $\log \hat{\mathcal{R}}_{\mathrm{rev}}^{-1}$ is a stochastic upper bound on $\log \mathcal{R} = \log \mathcal{R}_{\mathrm{rev}}^{-1}$. BDMC is

simply the combination of this stochastic upper bound with the stochastic lower bound of Section 2.2. Because AIS is a consistent estimator of the partition function ratio under the assumption of ergodicity [Nea01], the two bounds converge as $T \to \infty$; therefore, given enough computation, BDMC can sandwich $\log \mathcal{R}$ to arbitrary precision.

Returning to the setting of Bayesian inference, given some fixed observation $\mathbf{y}$, we can apply BDMC provided we have exact samples from both the prior distribution $p(\boldsymbol{\theta}, \mathbf{z})$ and the posterior distribution $p(\boldsymbol{\theta}, \mathbf{z}|\mathbf{y})$. In practice, the prior is typically easy to sample from, but it is typically infeasible to generate exact posterior samples. However, in models where we can tractably sample from the joint distribution $p(\boldsymbol{\theta}, \mathbf{z}, \mathbf{y})$, we can generate exact posterior samples for *simulated* observations using the elementary fact that

$$p(\mathbf{y}) \, p(\boldsymbol{\theta}, \mathbf{z}|\mathbf{y}) = p(\boldsymbol{\theta}, \mathbf{z}, \mathbf{y}) = p(\boldsymbol{\theta}, \mathbf{z}) \, p(\mathbf{y}|\boldsymbol{\theta}, \mathbf{z}). \tag{3}$$

In other words, if one ancestrally samples $\boldsymbol{\theta}$, $\mathbf{z}$, and $\mathbf{y}$, this is equivalent to first generating a dataset $\mathbf{y}$ and then sampling $(\boldsymbol{\theta}, \mathbf{z})$ exactly from the posterior. Therefore, for simulated data, one has access to a single exact posterior sample; this is enough to obtain stochastic upper bounds on $\log \mathcal{R} = \log p(\mathbf{y})$.

### 2.4 WebPPL and Stan

We focus on two particular probabilistic programming packages. First, we consider WebPPL [GS], a lightweight probabilistic programming language built on Javascript, and intended largely to illustrate some of the important ideas in probabilistic programming. Inference is based on Metropolis–Hastings (M–H) updates to a program's execution trace, i.e. a record of all stochastic decisions made by the program. WebPPL has a small and clean implementation, and the entire implementation is described in an online tutorial on probabilistic programming [GS].

Second, we consider Stan [CGHL+ p], a highly engineered automatic inference system which is widely used by statisticians and is intended to scale to large problems. Stan is based on the No U-Turn Sampler (NUTS; [HG14]), a variant of Hamiltonian Monte Carlo (HMC; [Nea+11]) which chooses trajectory lengths adaptively. HMC can be significantly more efficient than M–H over execution traces because it uses gradient information to simultaneously update multiple parameters of a model, but is less general because it requires a differentiable likelihood. (In particular, this disallows discrete latent variables unless they are marginalized out analytically.)

## 3 Methods

There are at least two criteria we would desire from a sampling-based approximate inference algorithm in order that its samples be representative of the true posterior distribution: we would like the approximate distribution $q(\boldsymbol{\theta}, \mathbf{z}; \mathbf{y})$ to cover all the high-probability regions of the posterior $p(\boldsymbol{\theta}, \mathbf{z}|\mathbf{y})$, and we would like it to avoid placing probability mass in low-probability regions of the posterior. The former criterion motivates measuring the KL divergence $\mathrm{D}_{\mathrm{KL}}(p(\boldsymbol{\theta}, \mathbf{z}|\mathbf{y}) \, \| \, q(\boldsymbol{\theta}, \mathbf{z}; \mathbf{y}))$, and the latter criterion motivates measuring $\mathrm{D}_{\mathrm{KL}}(q(\boldsymbol{\theta}, \mathbf{z}; \mathbf{y}) \, \| \, p(\boldsymbol{\theta}, \mathbf{z}|\mathbf{y}))$. If we desire both simultaneously, this motivates paying attention to the Jeffreys divergence, defined as $\mathrm{D}_{\mathrm{J}}(q\|p) = \mathrm{D}_{\mathrm{KL}}(q\|p) + \mathrm{D}_{\mathrm{KL}}(p\|q)$.

In this section, we present Bounding Divergences with Reverse Annealing (BREAD), a technique for using BDMC to bound the Jeffreys divergence from the true posterior on simulated data, combined with a protocol for using this technique to analyze sampler accuracy on real-world data.

### 3.1 Upper bounding the Jeffreys divergence in expectation

We now present our technique for bounding the Jeffreys divergence between the target distribution and the distribution of approximate samples produced by AIS. In describing the algorithm, we revert to the abstract state space formalism of Section 2.1, since the algorithm itself does not depend on any structure specific to posterior inference (except for the ability to obtain an exact sample). We first repeat the derivation from [GGA15] of the bias of the stochastic lower bound $\log \hat{\mathcal{R}}$. Let $\mathbf{v} = (\mathbf{x}_1, \ldots, \mathbf{x}_{T-1})$ denote all of the variables sampled in AIS *before the final stage*; the final state $\mathbf{x}_T$ corresponds to the approximate sample produced by AIS. We can write the distributions over the forward and reverse AIS chains as:

$$q_{fwd}(\mathbf{v}, \mathbf{x}_T) = q_{fwd}(\mathbf{v}) \, q_{fwd}(\mathbf{x}_T \,|\, \mathbf{v}) \tag{4}$$

$$q_{rev}(\mathbf{v}, \mathbf{x}_T) = p_T(\mathbf{x}_T) \, q_{rev}(\mathbf{v} \,|\, \mathbf{x}_T). \tag{5}$$

The distribution of approximate samples $q_{fwd}(\mathbf{x}_T)$ is obtained by marginalizing out $\mathbf{v}$. Note that sampling from $q_{rev}$ requires sampling exactly from $p_T$, so strictly speaking, BREAD is limited to those cases where one has at least one exact sample from $p_T$ — such as simulated data from a probabilistic model (see Section 2.3).

The expectation of the estimate $\log \hat{\mathcal{R}}$ of the log partition function ratio is given by:

$$\mathbb{E}[\log \hat{\mathcal{R}}] = \mathbb{E}_{q_{fwd}(\mathbf{v},\mathbf{x}_T)} \left[ \log \frac{f_T(\mathbf{x}_T)\, q_{rev}(\mathbf{v}\,|\,\mathbf{x}_T)}{\mathcal{Z}_1\, q_{fwd}(\mathbf{v},\mathbf{x}_T)} \right] \tag{6}$$

$$= \log \mathcal{Z}_T - \log \mathcal{Z}_1 - \mathrm{D_{KL}}(q_{fwd}(\mathbf{x}_T)\, q_{fwd}(\mathbf{v}\,|\,\mathbf{x}_T) \,\|\, p_T(\mathbf{x}_T)\, q_{rev}(\mathbf{v}\,|\,\mathbf{x}_T)) \tag{7}$$

$$\leq \log \mathcal{Z}_T - \log \mathcal{Z}_1 - \mathrm{D_{KL}}(q_{fwd}(\mathbf{x}_T) \,\|\, p_T(\mathbf{x}_T)). \tag{8}$$

(Note that $q_{fwd}(\mathbf{v}\,|\,\mathbf{x}_T)$ is the *conditional* distribution of the forward chain, given that the final state is $\mathbf{x}_T$.) The inequality follows because marginalizing out variables cannot increase the KL divergence.

We now go beyond the analysis in [GGA15], to bound the bias in the other direction. The expectation of the reverse estimate $\hat{\mathcal{R}}_{\mathrm{rev}}$ is

$$\mathbb{E}[\log \hat{\mathcal{R}}_{\mathrm{rev}}] = \mathbb{E}_{q_{rev}(\mathbf{x}_T,\mathbf{v})} \left[ \log \frac{\mathcal{Z}_1\, q_{fwd}(\mathbf{v},\mathbf{x}_T)}{f_T(\mathbf{x}_T)\, q_{rev}(\mathbf{v}\,|\,\mathbf{x}_T)} \right] \tag{9}$$

$$= \log \mathcal{Z}_1 - \log \mathcal{Z}_T - \mathrm{D_{KL}}(p_T(\mathbf{x}_T)\, q_{rev}(\mathbf{v}|\mathbf{x}_T) \,\|\, q_{fwd}(\mathbf{x}_T)\, q_{fwd}(\mathbf{v}|\mathbf{x}_T)) \tag{10}$$

$$\leq \log \mathcal{Z}_1 - \log \mathcal{Z}_T - \mathrm{D_{KL}}(p_T(\mathbf{x}_T) \,\|\, q_{fwd}(\mathbf{x}_T)). \tag{11}$$

As discussed above, $\log \hat{\mathcal{R}}$ and $\log \hat{\mathcal{R}}_{\mathrm{rev}}^{-1}$ can both be seen as estimators of $\log \frac{\mathcal{Z}_T}{\mathcal{Z}_1}$, the former of which is a stochastic lower bound, and the latter of which is a stochastic upper bound. Consider the gap between these two bounds, $\hat{\mathcal{B}} \triangleq \log \hat{\mathcal{R}}_{\mathrm{rev}}^{-1} - \log \hat{\mathcal{R}}$. It follows from Eqs. (8) and (11) that, in expectation, $\hat{\mathcal{B}}$ upper bounds the Jeffreys divergence

$$\mathcal{J} \triangleq \mathrm{D_J}(p_T(\mathbf{x}_T), q_{fwd}(\mathbf{x}_T)) \triangleq \mathrm{D_{KL}}(p_T(\mathbf{x}_T) \,\|\, q_{fwd}(\mathbf{x}_T)) + \mathrm{D_{KL}}(q_{fwd}(\mathbf{x}_T) \,\|\, p_T(\mathbf{x}_T)) \tag{12}$$

between the target distribution $p_T$ and the distribution $q_{fwd}(p_T)$ of approximate samples.

Alternatively, if one happens to have some other lower bound $\mathcal{L}$ or upper bound $\mathcal{U}$ on $\log \mathcal{R}$, then one can bound either of the one-sided KL divergences by running only one direction of AIS. Specifically, from Eq. (8), $\mathbb{E}[\mathcal{U} - \log \hat{\mathcal{R}}] \geq \mathrm{D_{KL}}(q_{fwd}(\mathbf{x}_T) \,\|\, p_T(\mathbf{x}_T))$, and from Eq. (11), $\mathbb{E}[\log \hat{\mathcal{R}}_{\mathrm{rev}}^{-1} - \mathcal{L}] \geq \mathrm{D_{KL}}(p_T(\mathbf{x}_T) \,\|\, q_{fwd}(\mathbf{x}_T))$.

How tight is the expectation $\mathcal{B} \triangleq \mathbb{E}[\hat{\mathcal{B}}]$ as an upper bound on $\mathcal{J}$? We evaluated both $\mathcal{B}$ and $\mathcal{J}$ exactly on some toy distributions and found them to be a fairly good match. Details are given in Appendix B.

### 3.2 Application to real-world data

So far, we have focused on the setting of simulated data, where it is possible to obtain an exact posterior sample, and then to rigorously bound the Jeffreys divergence using BDMC. However, we are more likely to be interested in evaluating the performance of inference on real-world data, so we would like to simulate data which resembles a real-world dataset of interest. One particular difficulty is that, in Bayesian analysis, hyperparameters are often assigned non-informative or weakly informative priors, in order to avoid biasing the inference. This poses a challenge for BREAD, as datasets generated from hyperparameters sampled from such priors (which are often very broad) can be very dissimilar to real datasets, and hence conclusions from the simulated data may not generalize.

In order to generate simulated datasets which better match a real-world dataset of interest, we adopt the following heuristic scheme: we first perform approximate posterior inference on the real-world dataset. Let $\hat{\boldsymbol{\eta}}_{\mathrm{real}}$ denote the estimated hyperparameters. We then simulate parameters and data from the forward model $p(\boldsymbol{\theta}\,|\,\hat{\boldsymbol{\eta}}_{\mathrm{real}})p(\mathcal{D}\,|\,\hat{\boldsymbol{\eta}}_{\mathrm{real}},\boldsymbol{\theta})$. The forward AIS chain is run on $\mathcal{D}$ in the usual way. However, to initialize the reverse chain, we first start with $(\hat{\boldsymbol{\eta}}_{\mathrm{real}}, \boldsymbol{\theta})$, and then run some number of MCMC transitions which preserve $p(\boldsymbol{\eta}, \boldsymbol{\theta}\,|\,\mathcal{D})$, yielding an approximate posterior sample $(\boldsymbol{\eta}^\star, \boldsymbol{\theta}^\star)$.

In general, $(\boldsymbol{\eta}^\star, \boldsymbol{\theta}^\star)$ will not be an exact posterior sample, since $\hat{\boldsymbol{\eta}}_{\mathrm{real}}$ was not sampled from $p(\boldsymbol{\eta}\,|\,\mathcal{D})$. However, the true hyperparameters $\hat{\boldsymbol{\eta}}_{\mathrm{real}}$ which generated $\mathcal{D}$ ought to be in a region of high posterior mass unless the prior $p(\boldsymbol{\eta})$ concentrates most of its mass away from $\hat{\boldsymbol{\eta}}_{\mathrm{real}}$. Therefore, we expect even a small number of MCMC steps to produce a plausible posterior sample. This motivates our use of $(\boldsymbol{\eta}^\star, \boldsymbol{\theta}^\star)$ in place of an exact posterior sample. We validate this procedure in Section 5.1.2.

# 4  Related work

Much work has been devoted to the diagnosis of Markov chain convergence (e.g. [CC96; GR92; BG98]). Diagnostics have been developed both for estimating the autocorrelation function of statistics of interest (which determines the number of effective samples from an MCMC chain) and for diagnosing whether Markov chains have reached equilibrium. In general, convergence diagnostics cannot confirm convergence; they can only identify particular forms of non-convergence. By contrast, BREAD can rigorously demonstrate convergence in the simulated data setting.

There has also been much interest in automatically configuring parameters of MCMC algorithms. Since it is hard to reliably summarize the performance of an MCMC algorithm, such automatic configuration methods typically rely on method-specific analyses. For instance, Roberts and Rosenthal [RR01] showed that the optimal acceptance rate of Metropolis–Hastings with an isotropic proposal distribution is 0.234 under fairly general conditions. M–H algorithms are sometimes tuned to achieve this acceptance rate, even in situations where the theoretical analysis doesn't hold. Rigorous convergence measures might enable more direct optimization of algorithmic hyperparameters.

Gorham and Mackey [GM15] presented a method for directly estimating the quality of a set of approximate samples, independently of how those samples were obtained. This method has strong guarantees under a strong convexity assumption. By contrast, BREAD makes no assumptions about the distribution itself, so its mathematical guarantees (for simulated data) are applicable even to multimodal or badly conditioned posteriors.

It has been observed that heating and cooling processes yield bounds on log-ratios of partition functions by way of finite difference approximations to thermodynamic integration. Neal [Nea96] used such an analysis to motivate tempered transitions, an MCMC algorithm based on heating and cooling a distribution. His analysis cannot be directly applied to measuring convergence, as it assumed equilibrium at each temperature. Jarzynski [Jar97] later gave a non-equilibrium analysis which is equivalent to that underlying AIS [Nea01].

We have recently learned of independent work [CTM16] which also builds on BDMC to evaluate the accuracy of posterior inference in a probabilistic programming language. In particular, Cusumano-Towner and Mansinghka [CTM16] define an unbiased estimator for a quantity called the *subjective divergence*. The estimator is *equivalent* to BDMC except that the reverse chain is initialized from an arbitrary *reference* distribution, rather than the true posterior. In [CTM16], the subjective divergence is shown to upper bound the Jeffreys divergence when the true posterior is used; this is equivalent to our analysis in Section 3.1. Much less is known about subjective divergence when the reference distribution is not taken to be the true posterior. (Our approximate sampling scheme for hyperparameters can be viewed as a kind of reference distribution.)

# 5  Experiments

In order to experiment with BREAD, we extended both WebPPL and Stan to run forward and reverse AIS using the sequence of distributions defined in Eq. (2). The MCMC transition kernels were the standard ones provided by both platforms. Our first set of experiments was intended to validate that BREAD can be used to evaluate the accuracy of posterior inference in realistic settings. Next, we used BREAD to explore the tradeoffs between two different representations of a matrix factorization model in both WebPPL and Stan.

## 5.1  Validation

As described above, BREAD returns rigorous bounds on the Jeffreys divergence only when the data are sampled from the model distribution. Here, we address three ways in which it could potentially give misleading results. First, the upper bound $\mathcal{B}$ may overestimate the true Jeffreys divergence $\mathcal{J}$. Second, results on simulated data may not correspond to results on real-world data if the simulated data are not representative of the real-world data. Finally, the fitted hyperparameter procedure of Section 3.2 may not yield a sample sufficiently representative of the true posterior $p(\boldsymbol{\theta}, \boldsymbol{\eta} | \mathcal{D})$. The first issue, about the accuracy of the bound, is addressed in Appendix B.1.1; the bound appears to be fairly close to the true Jeffreys divergence on some toy distributions. We address the other two issues in this section. In particular, we attempted to validate that the behavior of the method on simulated

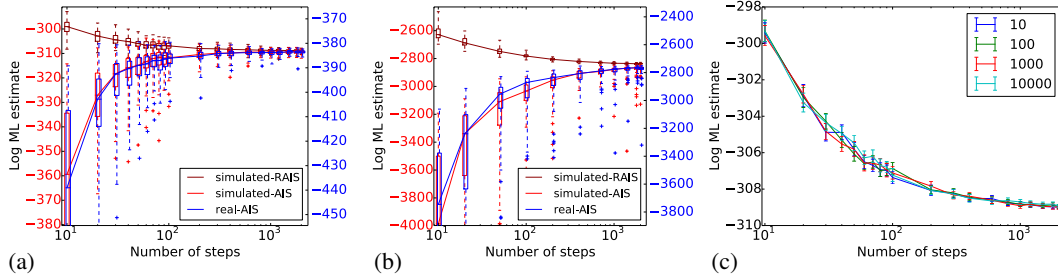

Figure 1: **(a)** Validation of the consistency of the behavior of forward AIS on real and simulated data for the logistic regression model. Since the log-ML values need not match between the real and simulated data, the y-axes for each curve are shifted based on the maximum log-ML lower bound obtained by forward AIS. **(b)** Same as (a), but for matrix factorization. The complete set of results on all datasets is given in Appendix D. **(c)** Validation of the fitted hyperparameter scheme on the logistic regression model (see Section 5.1.2 for details). Reverse AIS curves are shown as the number of Gibbs steps used to initialize the hyperparameters is varied.

data is consistent with that on real data, and that the fitted-hyperparameter samples can be used as a proxy for samples from the posterior. All experiments in this section were performed using Stan.

### 5.1.1 Validating consistency of inference behavior between real and simulated data

To validate BREAD in a realistic setting, we considered five models based on examples from the Stan manual [Sta], and chose a publicly available real-world dataset for each model. These models include: linear regression, logistic regression, matrix factorization, autoregressive time series modeling, and mixture-of-Gaussians clustering. See Appendix C for model details and Stan source code.

In order to validate the use of simulated data as a proxy for real data in the context of BREAD, we fit hyperparameters to the real-world datasets and simulated data from those hyperparameters, as described in Section 3.2. In Fig. 1 and Appendix D, we show the distributions of forward and reverse AIS estimates on simulated data and forward AIS estimates on real-world data, based on 100 AIS chains for each condition.[2] Because the distributions of AIS estimates included many outliers, we visualize quartiles of the estimates rather than means.[3] The real and simulated data need not have the same marginal likelihood, so the AIS estimates for real and simulated data are shifted vertically based on the largest forward AIS estimate obtained for each model. For all five models under consideration, the forward AIS curves were nearly identical (up to a vertical shift), and the distributions of AIS estimates were very similar at each number of AIS steps. (An example where the forward AIS curves failed to match up due to model misspecification is given in Appendix D.) Since the inference behavior appears to match closely between the real and simulated data, we conclude that data simulated using fitted hyperparameters can be a useful proxy for real data when evaluating inference algorithms.

### 5.1.2 Validating the approximate posterior over hyperparameters

As described in Section 3.2, when we simulate data from fitted hyperparameters, we use an approximate (rather than exact) posterior sample $(\boldsymbol{\eta}^\star, \boldsymbol{\theta}^\star)$ to initialize the reverse chain. Because of this, BREAD is not mathematically guaranteed to upper bound the Jeffreys divergence even on the simulated data. In order to determine the effect of this approximation in practice, we repeated the procedure of Section 5.1.1 for all five models, but varying $S$, the number of MCMC steps used to obtain $(\boldsymbol{\eta}^\star, \boldsymbol{\theta}^\star)$, with $S \in \{10, 100, 1000, 10000\}$. The reverse AIS estimates are shown in Fig. 1 and Appendix D. (We do not show the forward AIS estimates because these are unaffected by $S$.) In all five cases, the reverse AIS curves were statistically indistinguishable. This validates our use of fitted hyperparameters, as it suggests that the use of approximate samples of hyperparameters has little impact on the reverse AIS upper bounds.

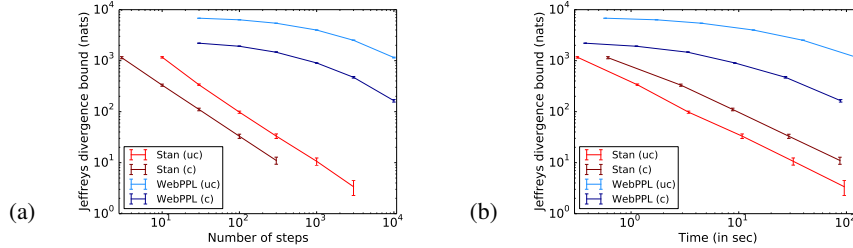

Figure 2: Comparison of Jeffreys divergence bounds for matrix factorization in Stan and WebPPL, using the collapsed and uncollapsed formulations. Given as a function of **(a)** number of MCMC steps, **(b)** running time.

## 5.2 Scientific findings produced by BREAD

Having validated various aspects of BREAD, we applied it to investigate the choice of model representation in Stan and WebPPL. During our investigation, we also uncovered a bug in WebPPL, indicating the potential usefulness of BREAD as a means of testing the correctness of an implementation.

### 5.2.1 Comparing model representations

Many models can be written in more than one way, for example by introducing or collapsing latent variables. Performance of probabilistic programming languages can be sensitive to such choices of representation, and the representation which gives the best performance may vary from one language to another. We consider the matrix factorization model described above, which we now specify in more detail. We approximate an $N \times D$ matrix $\mathbf{Y}$ as a low rank matrix, the product of matrices $\mathbf{U}$ and $\mathbf{V}$ with dimensions $N \times K$ and $K \times D$ respectively (where $K < \min(N, D)$). We use a spherical Gaussian observation model, and spherical Gaussian priors on $\mathbf{U}$ and $\mathbf{V}$:

$$u_{ik} \sim \mathcal{N}(0, \sigma_u^2) \qquad v_{kj} \sim \mathcal{N}(0, \sigma_v^2) \qquad y_{ij} \mid \mathbf{u}_i, \mathbf{v}_j \sim \mathcal{N}(\mathbf{u}_i^\top \mathbf{v}_j, \sigma^2)$$

We can also collapse $\mathbf{U}$ to obtain the model $v_{kj} \sim \mathcal{N}(0, \sigma_v^2)$ and $\mathbf{y}_i \mid \mathbf{V} \sim \mathcal{N}(\mathbf{0}, \sigma_u \mathbf{V}^\top \mathbf{V} + \sigma \mathbf{I})$. In general, collapsing variables can help MCMC samplers mix faster at the expense of greater computational cost per update. The precise tradeoff can depend on the size of the model and dataset, the choice of MCMC algorithm, and the underlying implementation, so it would be useful to have a quantitative criterion to choose between them.

We fixed the values of all hyperparameters to 1, and set $N = 50$, $K = 5$ and $D = 25$. We ran BREAD on both platforms (Stan and WebPPL) and for both formulations (collapsed and uncollapsed) (see Fig. 2). The simulated data and exact posterior sample were shared between all conditions in order to make the results directly comparable.

As predicted, the collapsed sampler resulted in slower updates but faster convergence (in terms of the number of steps). However, the per-iteration convergence benefit of collapsing was much larger in WebPPL than in Stan (perhaps because of the different underlying inference algorithm). Overall, the tradeoff between efficiency and convergence speed appears to favour the uncollapsed version in Stan, and the collapsed version in WebPPL (see Fig. 2(b)). (Note that this result holds only for our particular choice of problem size; the tradeoff may change given different model or dataset sizes.) Hence BREAD can provide valuable insights into the tricky question of which representations of models to choose to achieve faster convergence.

### 5.2.2 Debugging

Mathematically, the forward and reverse AIS chains yield lower and upper bounds on $\log p(\mathbf{y})$ with high probability; if this behavior is not observed, that indicates a bug. In our experimentation with WebPPL, we observed a case where the reverse AIS chain yielded estimates significantly lower than those produced by the forward chain, inconsistent with the theoretical guarantee. This led us to find a subtle bug in how WebPPL sampled from a multivariate Gaussian distribution (which had the effect that the exact posterior samples used to initialize the reverse chain were incorrect).[4] These days, while many new probabilistic programming languages are emerging and many are in active development, such debugging capabilities provided by BREAD can potentially be very useful.

## Footnotes

[1]Traditionally, this has meant having access to an exact sampler. However, in this work, we sometimes have access to a *sample* from $p_1$, but not a *sampler*.

[2]The forward AIS chains are independent, while the reverse chains share an initial state.

[3]Normally, such outliers are not a problem for AIS, because one averages the weights $w_T$, and this average is insensitive to extremely small values. Unfortunately, the analysis of Section 3.1 does not justify such averaging, so we report estimates corresponding to individual AIS chains.

[4]Issue: `https://github.com/probmods/webppl/issues/473`

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
