[Supplementary Material · supplemental.pdf]

## A Importance sampling analysis of AIS

This is well known and appears even in [Nea01]. We present it here in order to provide some foundation for the analysis of the bias of the log marginal likelihood estimators and the relationship to the Jeffreys divergence.

Neal [Nea01] justified AIS by showing that it is a simple importance sampler over an extended state space: To see this, note that the joint distribution of the sequence of states $\mathbf{x}_1, \ldots, \mathbf{x}_{T-1}$ visited by the algorithm has joint density

$$q_{fwd}(\mathbf{x}_1, \ldots, \mathbf{x}_{T-1}) = p_1(\mathbf{x}_1) \, \mathcal{T}_2(\mathbf{x}_2 | \mathbf{x}_1) \cdots \mathcal{T}_{T-1}(\mathbf{x}_{T-1} | \mathbf{x}_{T-2}), \tag{13}$$

and, further note that, by reversibility,

$$\mathcal{T}_t(\mathbf{x}' | \mathbf{x}) = \mathcal{T}_t(\mathbf{x} | \mathbf{x}') \frac{p_t(\mathbf{x}')}{p_t(\mathbf{x})} = \mathcal{T}_t(\mathbf{x} | \mathbf{x}') \frac{f_t(\mathbf{x}')}{f_t(\mathbf{x})}. \tag{14}$$

Then, applying Eq. (1)

$$\mathbb{E}[w] = \mathbb{E}\left[ \prod_{t=2}^{T} \frac{f_t(\mathbf{x}_{t-1})}{f_{t-1}(\mathbf{x}_{t-1})} \right] \tag{15}$$

$$= \mathbb{E}\left[ \frac{f_T(\mathbf{x}_{T-1})}{f_1(\mathbf{x}_1)} \frac{f_2(\mathbf{x}_1)}{f_2(\mathbf{x}_2)} \cdots \frac{f_{T-1}(\mathbf{x}_{T-2})}{f_{T-1}(\mathbf{x}_{T-1})} \right] \tag{16}$$

$$= \frac{\mathcal{Z}_T}{\mathcal{Z}_1} \mathbb{E}\left[ \frac{p_T(\mathbf{x}_{T-1})}{p_1(\mathbf{x}_1)} \frac{\mathcal{T}_2(\mathbf{x}_1 | \mathbf{x}_2)}{\mathcal{T}_2(\mathbf{x}_2 | \mathbf{x}_1)} \cdots \frac{\mathcal{T}_{T-1}(\mathbf{x}_{T-2} | \mathbf{x}_{T-1})}{\mathcal{T}_{T-1}(\mathbf{x}_{T-1} | \mathbf{x}_{T-2})} \right] \tag{17}$$

$$= \frac{\mathcal{Z}_T}{\mathcal{Z}_1} \mathbb{E}\left[ \frac{q_{rev}(\mathbf{x}_1, \ldots, \mathbf{x}_{T-1})}{q_{fwd}(\mathbf{x}_1, \ldots, \mathbf{x}_{T-1})} \right] \tag{18}$$

$$= \frac{\mathcal{Z}_T}{\mathcal{Z}_1}, \tag{19}$$

where

$$q_{rev}(\mathbf{x}_1, \ldots, \mathbf{x}_{T-1}) = p_T(\mathbf{x}_{T-1}) \, \mathcal{T}_{T-1}(\mathbf{x}_{T-2} | \mathbf{x}_{T-1}) \cdots \mathcal{T}_2(\mathbf{x}_1 | \mathbf{x}_2) \tag{20}$$

is the density of a hypothetical *reverse* chain, where $\mathbf{x}_{T-1}$ is first sampled exactly from the distribution $p_T$, and the transition operators are applied in the reverse order. (In practice, the intractability of $p_T$ generally prevents one from simulating the reverse chain.)

## B How tight is the Jeffreys divergence upper bound?

In Section 3.1, we derived an estimator $\hat{\mathcal{B}}$ which upper bounds in expectation the Jeffreys divergence $\mathcal{J}$ between the true posterior distribution and the distribution of approximate samples produced by AIS, i.e., $\mathbb{E}[\hat{\mathcal{B}}] > \mathcal{J}$. It can be seen from Eqs. (7) and (10) that the expectation $\mathcal{B} \triangleq \mathbb{E}[\hat{\mathcal{B}}]$ is the Jeffreys divergence between the distributions over the forward and reverse chains:

$$\begin{aligned} \mathcal{B} &= \mathbb{E}[\log \hat{\mathcal{R}}_{rev}^{-1} - \log \hat{\mathcal{R}}] \\ &= \mathrm{D}_{\mathrm{KL}}(q_{fwd}(\mathbf{x}_T) \, q_{fwd}(\mathbf{v} | \mathbf{x}_T) \,\|\, p_T(\mathbf{x}_T) \, q_{rev}(\mathbf{v} | \mathbf{x}_T)) + \\ &\quad + \mathrm{D}_{\mathrm{KL}}(p_T(\mathbf{x}_T) \, q_{rev}(\mathbf{v} | \mathbf{x}_T) \,\|\, q_{fwd}(\mathbf{x}_T) \, q_{fwd}(\mathbf{v} | \mathbf{x}_T)) \\ &= \mathrm{D}_{\mathrm{KL}}(q_{fwd}(\mathbf{v}, \mathbf{x}_T) \,\|\, q_{rev}(\mathbf{v}, \mathbf{x}_T)) + \mathrm{D}_{\mathrm{KL}}(q_{rev}(\mathbf{v}, \mathbf{x}_T) \,\|\, q_{fwd}(\mathbf{v}, \mathbf{x}_T)) \\ &= \mathrm{D}_{\mathrm{J}}(q_{fwd}(\mathbf{v}, \mathbf{x}_T) \,\|\, q_{rev}(\mathbf{v}, \mathbf{x}_T)) \end{aligned} \tag{21}$$

One might intuitively expect $\mathcal{B}$ to be a very loose upper bound on $\mathcal{J}$, as it is a divergence over a much larger domain (specifically, of size $|\mathcal{X}|^T$, compared with $|\mathcal{X}|$). In order to test this empirically, we have evaluated both $\mathcal{J}$ and $\mathcal{B}$ on several toy distributions for which both quantities can be tractably computed. We found the naïve intuition to be misleading — on some of these distributions, $\mathcal{B}$ is a good proxy for $\mathcal{J}$. In this section, we describe how to exactly compute $\mathcal{B}$ and $\mathcal{J}$ in small discrete spaces; experimental results are given in Appendix B.1.1

### B.1 Methods

Even when the domain $\mathcal{X}$ is a small discrete set, distributions over the extended state space are too large to represent explicitly. However, because both the forward and reverse chains are Markov chains, all of the marginal distributions $q_{fwd}(\mathbf{x}_t, \mathbf{x}_{t+1})$ and $q_{rev}(\mathbf{x}_t, \mathbf{x}_{t+1})$ can be computed using the forward pass of the Forward–Backward algorithm. The final-state divergence $\mathcal{J}$ can be computed directly from the marginals over

Figure 3: Comparing the true Jeffreys divergence $\mathcal{J}$ against our upper bound $\mathcal{B}$ for the toy distributions of Fig. 4. **Solid lines:** $\mathcal{J}$. **Dashed lines:** $\mathcal{B}$.

Figure 4: Visualizations of the toy distributions used for analyzing the tightness of our bound. **Columns 1-3:** intermediate distributions for various inverse temperatures $\beta$. **Column 4:** target distribution. **Column 5:** Distribution of approximate samples produced by AIS with 100 intermediate distributions.

$\mathbf{x}_T$. The KL divergence $\mathrm{D_{KL}}(q_{fwd} \| q_{rev})$ can be computed as:

$$\mathrm{D_{KL}}(q_{fwd} \| q_{rev}) = \mathbb{E}_{q_{fwd}(\mathbf{x}_1,\dots,\mathbf{x}_T)} \left[ \log q_{fwd}(\mathbf{x}_1,\dots,\mathbf{x}_T) - \log q_{rev}(\mathbf{x}_1,\dots,\mathbf{x}_T) \right] \tag{22}$$

$$= \mathbb{E}_{q_{fwd}(\mathbf{x}_1)} \left[ \log q_{fwd}(\mathbf{x}_1) - \log q_{rev}(\mathbf{x}_1) \right] +$$

$$+ \sum_{t=1}^{T-1} \mathbb{E}_{q_{fwd}(\mathbf{x}_t,\mathbf{x}_{t+1})} \left[ \log q_{fwd}(\mathbf{x}_{t+1} | \mathbf{x}_t) - \log q_{rev}(\mathbf{x}_{t+1} | \mathbf{x}_t) \right], \tag{23}$$

and analogously for $\mathrm{D_{KL}}(q_{rev} \| q_{fwd})$. $\mathcal{B}$ is the sum of these quantities.

### B.1.1 Experiments

To validate the upper bound $\mathcal{B}$ as a measure of the accuracy of an approximate inference engine, we must first check that it accurately reflects the true Jeffreys divergence $\mathcal{J}$. This is hard to do in realistic settings since $\mathcal{J}$ is generally unavailable, but we can evaluate both quantities on small toy distributions using the technique of Appendix B.1.

We considered several toy distributions, where the domain $\mathcal{X}$ was taken to be a $7 \times 7$ grid. In all cases, the MCMC transition operator was Metropolis-Hastings, where the proposal distribution was uniform over moving one step in the four coordinate directions. (Moves which landed outside the grid were rejected.) For the intermediate distributions, we used geometric averages with a linear schedule for $\beta$.

First, we generated random unnormalized target distributions $f(x) = \exp(g(x))$, where each entry of $g$ was sampled independently from the normal distribution $\mathcal{N}(0,\sigma)$. When $\sigma$ is small, the M-H sampler is able to explore the space quickly, because most states have similar probabilities, and therefore most M-H proposals are accepted. However, when $\sigma$ is large, the distribution fragments into separated modes, and it is slow to move from one mode to another using a local M-H kernel. We sampled random target distributions using $\sigma = 2$ (which we

refer to as EASY RANDOM) and $\sigma = 10$ (which we refer to as HARD RANDOM). The target distributions, as well as some of the intermediate distributions, are shown in Fig. 4.

The other toy distribution we considered consists of four modes separated by a deep energy barrier. The unnormalized target distribution is defined by:

$$f(x) = \begin{cases} e^3 & \text{in the upper right quadrant} \\ 1 & \text{in the other three quadrants} \\ e^{-10} & \text{on the barrier} \end{cases} \tag{24}$$

In the target distribution, the dominant mode makes up about $87\%$ of the probability mass. Fig. 4 shows the target function as well as some of the AIS intermediate distributions. This example illustrates a common phenomenon in AIS, whereby the distribution splits into different modes at some point in the annealing process, and then the relative probability mass of those modes changes. Because of the energy barrier, we refer to this example as BARRIER.

The distributions of approximate samples produced by AIS with $T = 100$ intermediate distributions are shown in Fig. 4. For EASY RANDOM, the approximate distribution is quite accurate, with $\mathcal{J} = 0.13$. However, for the other two examples, the probability mass is misallocated between different modes because the sampler has a hard time moving between them. Correspondingly, the Jeffreys divergences are $\mathcal{J} = 4.73$ and $\mathcal{J} = 1.65$ for HARD RANDOM and BARRIER, respectively.

How well are these quantities estimated by the upper bound $\mathcal{B}$? Fig. 3 shows both $\mathcal{J}$ and $\mathcal{B}$ for all three toy distributions and numbers of intermediate distributions varying from 10 to 100,000. In the case of EASY RANDOM, the bound is quite far off: for instance, with 1000 intermediate distributions, $\mathcal{J} \approx 0.00518$ and $\mathcal{B} \approx 0.0705$, which differ by more than a factor of 10. However, the bound is more accurate in the other two cases: for HARD RANDOM, $\mathcal{J} \approx 1.840$ and $\mathcal{B} \approx 2.309$, a relative error of 26%; for BARRIER, $\mathcal{J} \approx 1.085$ and $\mathcal{B} \approx 1.184$, a relative error of about 9%. In these cases, according to Fig. 3, the upper bound remains accurate across several orders of magnitude in the number of intermediate distributions and in the true divergence $\mathcal{J}$. Roughly speaking, it appears that most of the difference between the forward and reverse AIS chains is explained by the difference in distributions over the final state. Even the extremely conservative bound in the case of EASY RANDOM is potentially quite useful, as it still indicates that the inference algorithm is performing well, in contrast with the other two cases.

Overall, we conclude that on these toy distributions, the upper bound $\mathcal{B}$ is accurate enough to give meaningful information about $\mathcal{J}$. This motivates using $\mathcal{B}$ as a diagnostic for evaluating approximate inference algorithms when $\mathcal{J}$ is unavailable.

## C Models and datasets

- **Linear regression**. Targets are modeled as a linear function of input variables, plus spherical Gaussian noise. Regression weights are assigned a spherical Gaussian prior. Variances of both Gaussians are drawn from a broad inverse-gamma prior.

  *Dataset*: A standardized version of the NHEFS dataset[5], which consists of several health-related attributes for 1746 people. We pick 5 of them namely gender, age, race, years of smoking and whether they quit smoking to predict weight gain between 1971 and 1982.

  ```
  data {
          int N;
          int K;
          matrix[N,K] x;
          vector[N] y;
  }

  parameters {
          // Hyperparameters.
          real<lower=0> sigma_sq;
          real<lower=0> scale_sq;

          // Parameters.
          real alpha;
          vector[K] beta;
  }
  ```

```
model {
        // Sampling hyperparameters.
        sigma_sq ~ inv_gamma(1, 1);
        scale_sq ~ inv_gamma(1, 1);

        // Sampling parameters.
        alpha ~ normal(0, sqrt(scale_sq));
        for (k in 1:K)
                beta[k] ~ normal(0, sqrt(scale_sq));

        // Sampling data.
        {
                vector[N] mu;
                mu <- x * beta + alpha;
                for (i in 1:N)
                        y[i] ~ normal(mu[i], sqrt(sigma_sq));
        }
}
```

- **Logistic regression**. Targets are Bernoulli distributed with mean parameters which are log-linear functions of the inputs. Regression weights are sampled from a spherical Gaussian distribution whose variance is assigned a broad inverse-gamma prior.

  *Dataset*: A standardized version of the Pima Indian Diabetes Dataset[6] which contains 8 health-related attributes for 768 women of Pima Indian heritage such as age, body mass index, diastolic blood pressure, number of times pregnant etc. The task is to predict whether a woman has diabetes.

```
data {
        int N;
        int K;
        matrix[N,K] x;
        int y[N];
}

parameters {
        // Hyperparameters.
        real<lower=0> scale_sq;

        // Parameters.
        real alpha;
        vector[K] beta;
}

model {
        // Sampling hyperparameters.
        scale_sq ~ inv_gamma(1, 1);

        // Sampling parameters.
        alpha ~ normal(0, sqrt(scale_sq));
        for (k in 1:K)
                beta[k] ~ normal(0, sqrt(scale_sq));

        // Sampling data.
        {
                vector[N] mu;
                mu <- x * beta + alpha;
                for (i in 1:N)
                        y[i] ~ bernoulli(inv_logit(mu[i]));
        }
}
```

- **Matrix factorization**. The dataset, a fully or partially observed matrix, is modelled as a low-rank matrix (i.e. a product of two matrices), plus Gaussian noise. Elements of the factors are assigned spherical Gaussian priors. Variances of all Gaussians are drawn from broad inverse-gamma priors.

*Dataset*: A subset of the MovieLens 100k dataset[7], which consists of movie ratings given by a set of viewers on a scale from 1 to 5. We picked the 50 most engaged viewers and the 50 most viewed movies, and assume 5 latent dimensions. We normalized the ratings to have mean 0.

```
data {
        int N;
        int L;
        int D;
        matrix[N,D] Y;
}

parameters {
        // Hyperparameters.
        real<lower=0> scale_u_sq;
        real<lower=0> scale_v_sq;
        real<lower=0> sigma_sq;

        // Parameters.
        matrix[N,L] U;
        matrix[L,D] V;
}

model {
        matrix[N,D] UV;

        // Sampling hyperparameters.
        scale_u_sq ~ inv_gamma(1, 1);
        scale_v_sq ~ inv_gamma(1, 1);
        sigma_sq ~ inv_gamma(1, 1);

        // Sampling parameters.
        for (i in 1:N)
                for (j in 1:L)
                        U[i,j] ~ normal(0, sqrt(scale_u_sq));

        for (i in 1:L)
                for (j in 1:D)
                        V[i,j] ~ normal(0, sqrt(scale_v_sq));

        // Sampling data.
        UV <- U * V;
        for (i in 1:N)
                for (j in 1:D)
                        if (Y[i,j] > -999999)
                                Y[i,j] ~ normal(UV[i,j], sqrt(
                                    sigma_sq));
}
```

- **Time series modeling**. We use an autoregressive time series model of order one, where the next data point is modelled as a linear function of the previous data point, plus Gaussian noise. Since the autoregressive coefficient is likely to be close to 1, we assign it a Gaussian prior with mean 1 and a small variance. The bias term is assigned a standard Gaussian prior, and the noise variance is assigned a broad inverse-gamma prior. All three are treated as hyperparameters.

  *Dataset*: Daily stock prices of Google, Inc. from 2-7-2005 to 7-7-2005[8], with 105 time steps in all.

```
data {
        int<lower=0> N;
        real y[N];
}

parameters {
  // Hyperparameters.
```

```
        real alpha;
    real beta;
    real<lower=0> sigma_sq;

    // Parameters.
}

model {
    // Sampling hyperparameters.
    alpha ~ normal(0, 1);
    beta ~ normal(1, 0.1);
    sigma_sq ~ inv_gamma(1, 1);

    // Sampling parameters.

    // Sampling data.
    for (n in 2:N)
        y[n] ~ normal(alpha + beta * y[n-1], sqrt(sigma_sq));
}
```

- **Clustering**. A mixture of finite Gaussians (one for each cluster), plus Gaussian noise. Mixture means are assigned spherical Gaussian priors and the mixing proportions are assigned a flat Dirichlet prior. The inter- and intra- cluster variances are drawn from broad inverse-gamma priors.

  *Dataset*: A standardized version of the Old-Faithful dataset[9], which contains 272 pairs of waiting time between eruptions and the durations of eruption for the Old Faithful geyser in Yellowstone National Park, Wyoming, USA. We set the number of clusters to 2.

```
data {
    int<lower=0> N;
    int<lower=0> D;
    int<lower=0> K;
    vector[D] y[N];
}

transformed data {
    real HALF_D_LOG_TWO_PI;
    vector[K] ONES_VECTOR;

    HALF_D_LOG_TWO_PI <- 0.5 * D * log(2 * 3.14159);
    for (k in 1:K)
        ONES_VECTOR[k] <- 1;
}

parameters {
    // Hyperparameters.
    real<lower=0> scale_sq;
    real<lower=0> sigma_sq;

    // Parameters.
    vector[D] mu[K];
    simplex[K] pi;
}

transformed parameters {
    real z[N,K];
    for (n in 1:N)
        for (k in 1:K)
            z[n, k] <- log(pi[k]) - HALF_D_LOG_TWO_PI - 0.5 * D * log(
                sigma_sq) - 0.5 * (dot_self(mu[k] - y[n]) / sigma_sq);
}

model {
    // Sampling hyperparameters.
```

```
    scale_sq ~ inv_gamma(1, 1);
    sigma_sq ~ inv_gamma(1, 1);

    // Sampling parameters.
    for (k in 1:K)
      for (d in 1:D)
        mu[k,d] ~ normal(0, sqrt(scale_sq));
    pi ~ dirichlet(ONES_VECTOR);

    // Sampling data.
    for (n in 1:N)
      increment_log_prob(log_sum_exp(z[n]));
}
```

# D   Full experimental results

Here we report the full set of experimental results on all models we considered. See Section 5.1 for descriptions of the experimental protocols.

Our complete set of results for real-data validation, shown in Fig. 5, includes one case where the forward AIS curves failed to line up due to model misspecification. In particular, the first time we ran the experiment with the matrix factorization model on the Movielens dataset, we had not centered the ratings to be zero mean. This resulted in model misspecification, because the factorization model as we defined it did not include an intercept term. As shown in Fig. 5(e), the forward AIS curves failed to match. We then normalized the ratings to have zero mean and re-ran the experiment. As shown in Fig. 5(f), the forward AIS curves matched quite well after normalization. This is the only case where we observed a mismatch in inference behavior between real and simulated data.

Figure 5: Validation of the consistency of the behavior of forward AIS on real and simulated data. Since the log-ML values need not match between the real and simulated data, the y-axes for each curve are shifted based on the maximum log-ML lower bound obtained by forward AIS. In most cases, the forward AIS curves have a broadly similar shape, suggesting the simulated data is a good proxy for the real-world data. **(a)** linear regression **(b)** logistic regression **(c)** time series modeling **(d)** clustering **(e)** matrix factorization, with uncentered data **(f)** matrix factorization, with centered data

Figure 6: Validation of the fitted hyperparameter scheme (see Section 5.1.2 for details). Reverse AIS curves are shown as the number of Gibbs steps used to initialize the hyperparameters is varied. There appear to be no significant differences between the conditions. **(a)** linear regression **(b)** logistic regression **(c)** time series modeling **(d)** clustering **(e)** matrix factorization

## Footnotes

[5] http://cdn1.sph.harvard.edu/wp-content/uploads/sites/1268/2015/07/nhefs_book.xlsx

[6]https://archive.ics.uci.edu/ml/datasets/Pima+Indians+Diabetes

[7]http://grouplens.org/datasets/movielens/100k/

[8]https://onlinecourses.science.psu.edu/stat501/sites/onlinecourses. science.psu.edu.stat501/files/ch15/google_stock.txt

[9]`http://www.stat.cmu.edu/~larry/all-of-statistics/=data/faithful.dat`