[Reviews · NeurIPS 2016]

Reviewer 1

Summary

This paper proposes using bidirectional Monte Carlo to estimate the J-S divergence between an MCMC sampler and the target distribution of interest, with a particular eye towards debugging and evaluating MCMC implementations in PPLs.

Qualitative Assessment

I like the idea of using BDMC to diagnose failures of a sampler to converge—this idea seems potentially quite useful since existing MCMC diagnostics are indeed problematic. Two very relevant references are [1,2], which also test the ability of MCMC samplers by running them on data sampled from the prior. Section 3.1: I found this derivation very hard to follow. For one thing, q(.) is used before it's defined. From context, it seems that it represents the marginal distribution of the AIS sampler after T steps, but that should be made explicit. Things are slightly clearer if one reads the supplement, but the reader shouldn't need to read the supplement to understand the main text. In general, it would be nice to have a clear statement of how to apply the method from start to finish somewhere—the reader has to piece this together a bit. Minor notes: It's a little cluttered to break the unobserved variables into z and theta, when in a general setting like this there is no difference between z and theta---they always appear together, so why not eliminate z or theta? Line 165: missing close parentheses. [1] Geweke, J. (2004). Getting it right: Joint distribution tests of posterior simulators. Journal of the American Statistical Association, 99(467), 799–804. http://doi.org/10.1198/016214504000001132 [2] Cook, S. R., Gelman, A., & Rubin, D. B. (2012). Validation of software for Bayesian models using posterior quantiles. Journal of Computational and Graphical Statistics, 15(3), 675–692. http://doi.org/10.1198/106186006X136976

Confidence in this Review

2-Confident (read it all; understood it all reasonably well)


Reviewer 2

Summary

This paper uses the recent bidirectioanl Monte Carlo to evaluate MCMC-based posterior inference methods. The authors analyze lower bounds of KL-divergence between the true posterior and the approximate distribution. The method is integrated into WebPPL and Stan.

Qualitative Assessment

Overall, this is a nice paper. The authors use a recent art, i.e., bidirectioanl Monte Carlo, to analyze the MCMC-based posterior inference. This is a hot problem and the idea is interesing. I recommend to accept this paper.

Confidence in this Review

2-Confident (read it all; understood it all reasonably well)


Reviewer 3

Summary

The authors propose a generic method for evaluation of MCMC transition operators based on a recently proposed method known as bidirectional Monte Carlo (BDMC). The BDMC method performs two annealed importance sampling (AIS) sweeps, which are essentially the same as those used in the Tempered Transitions (TT) method [Neal Stat & Comp 1996], but reinterprets the accumulated probabilities as a lower and upper bound on the marginal likelihood. This paper proposes to use the gap between these two bounds as a generic performance measure for MCMC transition operators. The presented technical contributions are implementations of BDMC for the Stan and WebPPL probabilistic programming systems, and a protocol for evaluation of quality of posterior inference.

Qualitative Assessment

This paper has some strong points and some not so strong points. The main strong point is that using BDMC to assess convergence of MCMC operators is a beautifully simple idea, and easy to implement, which in my opinion means that this work is potentially high impact. This is particularly true in the context of probabilistic programming systems, which indeed are the envisioned use case here, and I think all such systems would do well to at least implement this method. My main point of concern lie with the novelty and the depth of the technical contribution. The authors cite an arxiv submission on BDMC as existing work, but (I think wisely) choose to devote a relatively large amount of space to reiterating its description. Unfortunately this does mean that the main technical contributions presented in sections 3.1 and 3.2 are somewhat rushed, and it is unfortunately also here where the writing quality slips a bit. As a result some issues of clarity arise (see below). Clarity issues aside, the technical contributions in this paper are somewhat modest. Relative to the existing work on BDMC, the main novelty appears to the realization that the difference between the lower and upper bound is itself provides an upper bound on the Jensen-Shannon divergence. This is indeed a very nice observation, though arguably not an entirely novel once. This brings me a point about related work. The authors, somewhat curiously, do not cite Radford Neal's work on Tempered Transitions [Statistics and Computing (1996) 6, 353-366], which is highly relevant in this context. This citation is also missing from the manuscript on BDMC, which makes me wonder whether the authors are in fact aware of it. If so, this is unfortunate in the sense that there is a missed opportunity in pointing out connections between the two methods. TT performs the same reverse "heating" and forward "annealing" sweep as BDMC, but with the interest of a "tempering" MCMC transtion move to improve mixing in multimodal distributions. In TT the final sample from the reverse sweep is used as the initial sample for the forward sweep, and the final sample from the forward sweep is then used as a proposal which is accepted or rejected according to the ratio of importance weights accumulated in both sweeps. The acceptance ratio in TT is, as far a I can tell, precisely the ratio of the stochastic lower and upper bound. Section 3.3 of Neal's paper in fact presents an interpretation of this acceptance ratio in terms of two thermodynamic integration estimates that bound log Z_T - log Z_1 from above and below, noting that a TT sampler will accept with probability 1 when the two converge. The analogy with TT also leads me to wonder why the authors do spend more attention on the choice of β_t values. In TT it is well known that too large steps in Δβ = β_{t+1}-β_t will lead to a poor acceptance ratio, wheras too small steps Δβ waste computation. The ratio of the stochastic lower and upper bound in BDMC will also depend on Δβ and this provides some very clear intuitions. If your two bounds are not close enough, you need to rerun BDMC with smaller Δβ. Similarly, I suspect that comparisons between MCMC transition operators are likely to be unreliable when the gap between the log upper and lower bounds is larger >1 nat. When this is the case, one should again lower Δβ. Ιt hasn't quite become clear to me from reading the paper what the "auto" part in AutoBDMC is, but I think there is an opportunity here to come up with some schedule for iteratively decreasing Δβ until sufficiently reliable inference results are obtained. In this respect the experiments could be a little more thorough. It would be good to see a systematic study of how the lower and upper bound vary with the step size. These criticisms aside, I think this is not a bad paper. As noted above, I think the method proposed by the authors is potentially quite useful. Depending on whether the author's response can convincingly argue that issues with clarity and discussion of related work can be addressed by camera ready, I could be inclined to argue for acceptance. Clarity - The definition of $\vec{v}$ as "all of the variables sampled in AIS before the final stage" is rather loose. The authors presumably mean ${z_{1:T-1}, \theta_{1:T-1}$, but this is not immediately unambiguous since there are two AIS sweeps in the algorithm. This is not helped by the use of inconsistent labels "rev" and "fwd" on the one hand and "for" and "back" on the other (is this difference significant, or a mere accident?) - The authors do not define what estimator \hat{p}(y) they are taking the expectation of in equations (5-7) and (9-11), which makes it difficult to follow how these equations arise. Do the authors mean to write E[log \hat{R}_{rev}] and [log \hat{R}_{fwd}]? - Equation (6), as written, makes no sense (the distributions inside the KL divergence are identical). Based on equations (4) and (10), I take it the right terms should read something like p(z, θ | y) q_back(v | z, θ, y)? - As far as I can tell the authors never unambiguously define what they mean by AutoBDMC. This is presumably means (a) caculating the difference between the stochastic lower and upper bound in order to assess convergence and (b) using the fixed hyperparameter scheme described in section 3.2? Experiments - It is now clear how β_t is chosen. Looking at the code in the appendix I see `var step = 1/options.steps`. I assume that AutoBDMC is therefore parameterized by the number of steps T to go from β_T = 1 to β_1 = 1 / T? It would be helpful to state this somewhere. In figures 2 and 3, does "HMC / No-U-Turn steps" indicate that BDMC is performed with this number of steps? - The acronym RAIS, which presumably stands for reverse AIS, is not defined in the caption, nor are the variants (u)c(R)AIS, which I am guessing refer to (un)collapsed variants? - It would be helpful to increase the size of the axes and tick labels. Typos - Two closing parens in the D_KL terms are missing in the expression for D_JS - In the NUTS citation: Homan -> Hoffman

Confidence in this Review

3-Expert (read the paper in detail, know the area, quite certain of my opinion)


Reviewer 4

Summary

This paper presents a method--termed AutoBDMC--for assessing MCMC convergence. The method uses the recently-developed Bidirectional Monte Carlo method to compute stochastic upper and lower bounds for the log marginal likelihood (this relies on annealed importance sampling). The authors show that the gap between the upper and lower bounds is itself a bound on the Jensen-Shannon divergence between the true posterior and the distribution of approximate samples, thus providing a single scalar quantity to assess MCMC convergence behavior. Since the method can only analyze convergence on synthetic data generated from the joint distribution, the authors provide experimental evidence that its analyses on such synthetic data is predictive of behavior on real world data. They also demonstrate uses of their method for model selection and debugging purposes.

Qualitative Assessment

Overall, I found this paper quite readable and well-motivated. The presented method seems clearly useful, and the experimental evaluation is reasonably compelling. I found all of Section 3 harder to follow than the rest of the paper. The transition into 3.1 is especially jarring, as it launches into variable and equation definitions without motivating them. If the divergence bound in Section 3.1 is a new result (which I believe it is), then this should be made explicit, as the novelty/contribution of the paper seems to hinge on this. What, precisely, *is* AutoBDMC? This is never explicitly stated anywhere in the paper. In my understanding, it *seems* like AutoBDMC is the approach of using AIS and rAIS to analyze MCMC convergence. I would expect to see a statement to that effect. By the time Section 3.2 states that “…this poses a challenge for AutoBDMC,” AutoBDMC should be clearly defined. Some other minutiae: Line 149: Should p(theta, z | y) be p(theta, z, y)? Notational inconsistencies: the paper switches between using y and D to denote data/observations, which I found a bit confusing at times. WebPPL code in supplemental: Should be accompanied by (a) which version(s) of WebPPL this is known to work with, and (b) how to integrate into the WebPPL distribution in order to be used. Or even better, release this as a WebPPL package that can be installed via npm. If possible, a similar release for the Stan version of the code would also be much appreciated.

Confidence in this Review

2-Confident (read it all; understood it all reasonably well)


Reviewer 5

Summary

The authors consider the problem of assessing the reliability of MCMC outputs and comparing the efficiency of MCMC schemes. They propose to do so by using the upper and lower stochastic bounds on the log marginal likelihood given by the Bidirectional Monte Carlo (BMC) procedure. The idea is for example to choose, among available MCMC schemes, the one that produces the tightest BMC stochastic bounds given the same computational effort. The main challenge is that BMC requires an exact sample from the target measure to produce a valid stochastic upper bound. The authors propose some heuristics to overcome this issue. Their solution, in the context of Bayesian posterior sampling, involves performing the diagnostic on the posterior generated by synthetic data rather than actual data, leveraging the fact that for such posterior it is easier to produce good approximate samples. Finally the proposed methodology is tested on some examples.

Qualitative Assessment

The idea of using stochastic lower bounds given by Bidirectional Monte Carlo to perform MCMC convergence diagnostic is interesting. Even if the methodology seems to be specific to AIS and SMC samplers schemes and to produce diagnostic based only on the marginal likelihood, it could still be valuable as it is an attempt to have theoretically justified MCMC diagnostic tools. However the implementation has various drawbacks that, in my opinion, make the proposed diagnostic methodology not so appealing. Such drawbacks arise from the heuristic methodology attempting to overcome the fact that Bidirectional Monte Carlo needs an exact sample from the posterior to be valid. I mention three of them. - First the authors assume to know in advance (e.g. from previously run MCMC) a value of the hyperparameters, \eta_{real}, which is representative of the whole posterior. This assumes both that the algorithm used to find \eta_{real} had converged (which is basically the original problem of interest) and that the posterior can be well summarized with a single value. Both assumptions are strong and unjustified in general. - Then they assume that the posterior induced by synthetic data generated from \eta_{real} has properties (in terms of MCMC convergence) similar to the original posterior. This again is an unjustified assumption that, for example, should be carefully justified in contexts involving model mis-specifications. It seems that the authors try to overcome the difficulty posed by bidirectional monte carlo at the price of "changing the problem", i.e. performing diagnostic on a different posterior. - Then they adopt the optimistic view that starting from \eta_{real} allows the MCMC algorithm of interest to be quickly able to draw an almost exact sample from the posterior (which may be the case but again does not need to be). Therefore the proposed methodology seems to rely on various assumptions which, crucially, are closely related to the actual thing we wanted to assess in the first instance (the convergence properties of the original MCMC algorithm). I appreciate the fact that Section 5.1 shows some simple examples where these issues appear not to affect the conclusions drawn from the proposed methodology. Nevertheless the few simple examples are not convincing enough to claim that the various issues that the authors themselves recognize will not be critical in other, maybe more complicated, examples. The authors criticize previously proposed convergence diagnostic tools because they can fail in multimodal cases or in cases where few summary statistics are not sufficiently expressive (lines 203-205). I think the authors should explain why their method can do better in such scenarios and provide some evidences. More generally I think it would help to have simulations to compare their method with commonly used diagnostic methods (e.g. the ones described in sections 4) to show scenarios where it produces more reliable diagnostics.

Confidence in this Review

2-Confident (read it all; understood it all reasonably well)


Reviewer 6

Summary

The paper summarizes previoulsy published results on bidirectional Monte Carlo and reports on its application using two probabilstic programming languages.

Qualitative Assessment

There seem to be no original ideas in this paper, which describes the implementation of known techniques in WebPPL and Stan languages. Perhaps an MCMC workshop might be a more appropriate venue for this work.

Confidence in this Review

2-Confident (read it all; understood it all reasonably well)